# Prenatal Alcohol Exposure Disrupts CXCL16 Expression in Rat Hippocampus: Temporal and Sex Differences

**DOI:** 10.3390/ijms26051920

**Published:** 2025-02-23

**Authors:** Mayra Madeleine Padilla-Valdez, Margarita Belem Santana-Bejarano, Marisol Godínez-Rubí, Daniel Ortuño-Sahagún, Argelia Esperanza Rojas-Mayorquín

**Affiliations:** 1Laboratorio de Neuroinmunobiología Molecular, Instituto de Investigación en Ciencias Biomédicas (IICB), Centro Universitario de Ciencias de la Salud, Universidad de Guadalajara, Guadalajara C.P. 44340, Mexico; mayra.padilla2332@alumnos.udg.mx; 2Laboratorio de Patología Diagnóstica e Inmunohistoquímica, Centro de Investigación y Diagnóstico en Patología, Departamento de Microbiología y Patología, Centro Universitario de Ciencias de la Salud, Universidad de Guadalajara, Guadalajara C.P. 44340, Mexico; santana.bejarano@gmail.com (M.B.S.-B.); juliana.godinez@academicos.udg.mx (M.G.-R.); 3Departamento Materno Infantil, Centro Universitario de Tlajomulco, Universidad de Guadalajara, Tlajomulco C.P. 45641, Mexico

**Keywords:** alcohol, prenatal alcohol exposure, fetal alcohol spectrum disorder, FASD, neuroinflammation

## Abstract

Prenatal alcohol exposure (PAE) affects around 40,000 newborns every year and poses a significant health risk. Although much is already known about the neurotoxic mechanisms of PAE, new findings continue to emerge. Studies with mouse models show that PAE leads to overexpression of proinflammatory cytokines and chemokines in the brain, which disrupts important neurodevelopmental processes such as cell migration, survival and proliferation of neurons. The chemokine CXCL16 is overexpressed in the brain following various impairments, including PAE. This study shows that CXCL16 expression varies by developmental stage and sex, consistent with known sexual dimorphism in immune responses. In females, CXCL16 expression may be influenced by estrogen-related mechanisms, possibly related to the alcohol-mediated rebound effect described here. In contrast, the male hippocampus shows greater resilience to PAE-induced CXCL16 changes. Furthermore, the presence of CXCL16 in neuronal nuclei suggests a role in gene regulation, similar to other chemokines such as CCL5 and CXCL4. These findings shed light on the role of chemokines in hippocampal neuroplasticity and may pave the way for better treatment of fetal alcohol spectrum disorder (FASD).

## 1. Introduction

Prenatal alcohol exposure (PAE) is a serious health concern that affects approximately 40,000 newborns each year in the United States alone [1]. The consequences of this exposure cause a variety of neurological signs and symptoms ranging from mild behavioral disorders (e.g., attention deficits or learning problems) to severe malformations such as agenesis of the corpus callosum [2] and are grouped under the umbrella term fetal alcohol spectrum disorder (FASD), a disorder that can persist throughout life [3]. Although some of the neurotoxic mechanisms of PAE have been extensively studied, new mechanisms continue to emerge [4]. The role of neuroinflammation in PAE has been shown to underline some neurological defects that also occur in other neurodevelopmental disorders such as autism and schizophrenia [5]. In particular, murine models have shown that PAE causes overexpression of proinflammatory cytokines and chemokines (chemoattractant cytokines) in the brain, including interferon (IFN)-ɣ, interleukin (IL)-1β and tumor necrosis factor (TNF)-α [6,7]. In children, PAE increases serum levels of the chemokines eotaxin (CCL11), eotaxin-3 (CCL26) and basic fibroblast growth factor (bFGF) while causing a specific profile of other cytokines that are associated with neurodevelopmental delay in the same children: inhibition of macrophage inflammatory protein (MIP-1β, also known as CCL4), macrophage-derived chemokine (MDC or CCL22) and monocyte chemoattractant protein (MCP-4 or CCL13) and activation of C-reactive protein (CRP) [8]. Furthermore, neuroinflammation is not limited to early development. New findings show that an abnormal inflammatory response in the brain persists into adulthood. The hippocampus is one of the areas most affected by the effects of alcohol. In fact, PAE has been reported to decrease the volume and number of neurons in various subfields of the rat hippocampus [9], and proinflammatory chemokines such as monocyte chemoattractant protein 1 (MCP-1) are also increased in the hippocampus after PAE [10].

Chemokines are not only immune molecules, but they also orchestrate key neurodevelopmental processes, including cellular migration, survival and proliferation of neurons [11], all of which are disrupted by PAE. Similarly, chemokines have gained recognition under this paradigm and are currently considered the third most important communication system in the brain [12]. Of particular interest is the chemokine with the motif CXC ligand 16 (CXCL16), a scavenger receptor that was first described on the surface of macrophages in atheroma lesions [13]. Later, CXCL16 was found in endothelial cells, reactive astroglial cells, and glioma cells, where it showed chemotaxis properties on cells carrying its only known receptor: CXCR6 [14]. Soon after, CXCL16 was detected in the central nervous system (CNS), where it was proposed as a marker for inflammation and atherosclerosis [15]. In 2008, it was found that CXCL16 also responds to inflammatory signals such as tumor necrosis factor (TNF)-α, interferon (IFN)-γ or interleukin (IL)-15 and that CXCL16 induces glial cell migration and invasion in vitro [16]. Following these discoveries, CXCL16 was detected in neurons, astrocytes and microglia in vitro [17], where it was found to exert neuroprotective effects in hippocampal cells. Later, the same group found that the administration of CXCL16 reduced the volume of an ischemic lesion [18], suggesting that CXCL16 may play a more important role in the CNS.

In the brain, CXCL16 is overexpressed after various insults, namely ischemia, trauma, cancer and other inflammatory conditions such as prenatal alcohol exposure (PAE), finding a link with the disruption of the blood–brain barrier in this condition [19]. On the other hand, CXCL16 levels were only measured during pregnancy or immediately after birth. However, CXCL16 appears to be underexpressed in male rats when measured at postnatal day 38 [20], suggesting a possible regulatory mechanism that has not yet been explored. Furthermore, the immune signature is different in female rats during PAE [6], and so far, none of the CXCL16 studies have included a female animal. According to the Human Protein Atlas, CXCL16 RNA is virtually ubiquitous, especially in the cortex and spinal cord [21]. Therefore, the aim of this study is to investigate the expression patterns of CXCL16 in the brain after prenatal alcohol exposure (PAE). In particular, this research aims to investigate the spatial distribution of the chemokine, assess the influence of sexual dimorphism on its expression and explore potential developmental changes over time. These results will improve our understanding of the presence and role of CXCL16 in the brain.

## 2. Results

### 2.1. PAE Affects CXCL16 Expression Differently in Males and Females

CXCL16 expression was not constant across developmental stages, sexes and hippocampal subfields. In CA1 of males (Figure 1), there was a statistically significant reduction in the EtOH group compared to the ISO group at PD 38 (*p* = 0.025). However, in this specific subfield, no difference was found between the groups and the age of females.

On the other hand, CA3 proved to be a particularly sensitive area in females, where differences were found in almost all observed age groups (Figure 2). There is an overexpression of CXCL16 in EtOH compared to ISO at PD21 (*p* = 0.0068), and such overexpression is still observed between the same groups at PD 38, although the effect is smaller (*p* = 0.037). Interestingly, the opposite effect is observed at PD 45, where a drastic decrease in CXCL16 expression is observed in the EtOH group compared to ISO (*p* = 0.017) and INT (*p* = 0.024). At PD 60, all group differences between females disappear. Despite the PAE effect on this subfield in females, no differences within groups were observed in their male counterparts.

As for the effect of PAE on the DG, there is a dimorphic response. In males, the EtOH group showed a decrease in CXCL16 expression at PD38 compared to the INT group (*p* = 0.039) and the ISO group (*p* = 0.030). In contrast, the PAE effect in females is the opposite and increases the CXCL16 at PD21 in the EtOH group to 93.47% of cells, which is a statistically significant difference compared to the ISO group (*p* = 0.015) but not compared to the INT group (Figure 3).

### 2.2. PAE Leads to Changes in CXCL16 Throughout Development

CXCL16 expression in the hippocampus varies at different developmental stages and appears to follow distinct patterns in males and females. In the female CA1 region (Figure 4), CXCL16 expression in the INT group starts at 85.61% at postnatal day (PD) 21 and decreases to a low of 53.18% by PD60. In contrast, the expression of CXCL16 in the ISO group remains relatively high and stable, ranging from 91.81% at PD21 to its lowest value at PD45 with 73.79%. On the other hand, the EtOH group starts with the highest expression of CXCL16 at PD21 with 93.46%, then decreases to 63.70% by PD38 and reaches its lowest value at PD45 with 32.91%, and then slightly increases again at PD60 with 63.12%. Overall, the expression of CXCL16 in female CA1 shows a sigmoidal trend in all groups, while expression appears to be impaired in the EtOH group, where CXCL16 expression drops dramatically during PD45 compared to the first age evaluated (*p* = 0.0053).

In males, CXCL16 expression in the CA1 region fluctuates across the ages analyzed but does not reach statistical significance. However, the EtOH group follows a similar trend to the female EtOH group, with CXCL16 levels decreasing to 55.91% by PD38. These results indicate that prenatal alcohol exposure (PAE) affects CXCL16 expression not only during in utero exposure but also its modulates it throughout development. This effect is particularly pronounced in females, where the changes are more pronounced.

The female CA3 region again showed a particular sensitivity to the effects of prenatal alcohol exposure (PAE). In the EtOH group, CXCL16 expression decreased significantly from PD21 to PD45 (*p* = 0.0006) and from PD38 to PD45 (*p* = 0.006). Expression increased up to PD60 compared to PD45 (*p* = 0.019).

When we compare these results with those observed in the female CA1 region, we see a similar pattern in the female CA3 of the EtOH group, which shows a significant decrease in CXCL16 expression at PD45 followed by an increase at PD60 (Figure 5). This trend is also observed in other regions, and similar patterns are observed in males in the EtOH groups. We refer to this effect as Alcohol-Mediated Rebound Response (AMRR). On the other hand, the expression of CXCL16 in CA3 was significantly lower in males only at PD38 compared to PD60 (*p* = 0.011).

Finally, CXCL16 expression in the male DG showed no differences between groups and age groups. In contrast, the EtOH group in females showed the AMRR pattern: CXCL16 expression started beginning at 97.52% at PD21, decreased to a similar level as in the ISO and INT groups by PD38, reached a low of 7.93% by PD45 and recovered to a level close to control values by PD60. Furthermore, the increased CXCL16 expression in the female EtOH group was statistically significant at PD21 compared to PD38 (*p* = 0.004), PD45 (*p* = 0.00021) and PD60 (*p* = 0.0032) (Figure 6).

It is important to note that the ISO and INT groups did not behave completely the same, as the observed differences between the EtOH-INT and EtOH-ISO comparisons were slightly different. This is an interesting fact considering that most studies on PAE only include an ISO group and no intact group as reference. This discrepancy could be related to the effects that sucrose consumption could have on some areas of the brain. In the hippocampus, for example, sucrose consumption has been associated with impaired spatial memory and increased neuroinflammatory markers such as IL-6 and IL-1β [22] and could also affect neurogenesis in a similar way to early childhood stress exposure [23]. However, in the case of CXCL16, the changes in the regulation of Cxcl16 expression in the brain are due to prenatal alcohol exposure rather than sucrose, as shown by our results here, as well as those previously described [20].

### 2.3. CXCL16 Is Mainly Expressed in the Neurons of the Brain

The DG is the first component of the hippocampus that relays information via mossy fibers to the CA3, which in turn is connected to the CA1 via Schaffer collaterals. Given this critical connectivity, we focused on understanding the distribution of CXCL16 in these areas.

This is particularly important as our previous results indicate that PAE consistently increases early CXCL16 expression in females compared to males at PD21. However, this increased expression could originate from neurons, microglia or astrocytes, so further investigation of cellular localization was required.

When comparing males and females exposed to EtOH in PD21, we found that CXCL16 expression in males was mainly localized in the soma and nucleus of neurons in the granule cell layer. In contrast, CXCL16 expression in females, was significantly increased on the apical side of granule cell neurons projecting to the inner molecular layer. This indicates not only an overproduction of the protein but also its mobilization within these cells (Figure 7).

When analyzing the DG, the colocalization of CXCL16 with NeuN indicates that most neurons were expressing CXCL16 throughout all the groups, along with some of the NeuN cells in the layer of the subgranular zone. In contrast, IBA-1/CXCL16 and GFAP/CXCL16 coexpression in the DG does not show the same pattern; in Figure 8, we show as an example the group with highest expression of CXCL16 (females from EtOH at PD21) showing that this chemokine might be primarily produced by neurons.

### 2.4. PAE Alters CXCL16 in the Cerebellum

Although it was out of the scope of this study, we also analyzed the overall expression of CXCL16 in different regions of the brain and found that the expression was predominantly localized in neurons. Interestingly, we observed a pronounced expression of CXCL16 in the Purkinje cells of the cerebellum, a region known for its sensitivity to alcohol.

Further analysis revealed that CXCL16 expression was generally similar in males and females. Moreover, Purkinje cells are among the few neurons that do not normally express the NeuN protein. Surprisingly, we observed a rare phenomenon in rats treated in the EtOH group at PD21, where their Purkinje cells stained positive for NeuN (Figure 9). This unusual observation could have implications for the functionality of these cells or could indicate a disruption in cell programming caused by EtOH exposure. Remarkably, this phenomenon was not observed at other developmental stages.

## 3. Discussion

In humans, PAE can affect the volume, organization and function of the hippocampus [24,25,26,27,28,29], a structure that plays an important role in memory retrieval and verbal learning. PAE is also known to affect the immune system, making its effects even more complex.

In this study, we have shown that the expression of CXCL16 varies at different developmental stages and shows differences between the sexes, which is consistent with the well-documented sexual dimorphism in immune responses. Sex is a critical factor for immune system milestones, such as microglial colonization in the rat brain. For example, females show delayed microglial colonization and distinct cytokine expression profiles, including variations in *Cxcl16* expression in the hippocampus depending on age (PD0, PD4 and PD60) and sex [30].

An intriguing phenomenon was observed in the female hippocampus at PD45, where CXCL16 levels decreased significantly in all subregions (CA1, CA3 and DG). This decrease coincides with the downregulation of other genes in the hypothalamus, such as Esr1 and Kiss1 [31], which are critical for estrogen regulation. These results suggest that the expression of CXCL16 in females may be influenced by estrogen-related mechanisms, possibly related to the Alcohol-Mediated Rebound Response (AMRR) we identified.

In contrast, the male hippocampus showed greater resilience to PAE-induced CXCL16 alterations. Remarkably, males showed an AMRR effect at PD38 and returned to normal levels at PD60, just like their female counterparts. Immune responses tend to be less pronounced in males [32]. However, it remains unclear whether this difference is an advantage or disadvantage in the context of PAE and CXCL16.

CXCL16 expression has previously been studied in vitro, where it was found in most central nervous system cell types [18]. However, our in vivo analysis revealed that CXCL16 expression is mainly localized in neurons in the hippocampus, suggesting a possible role in neuron–glia communication, as previously noted by other groups [18]. In addition, CXCL16 was also detected in neuronal nuclei, suggesting a possible role in gene regulation. This nuclear localization is consistent with the findings of other chemokines such as CCL5 and CXCL4, which are translocated to the nucleus in endothelial cells [33].

According to the Human Protein Atlas [34] (available at v19.proteinatlas.org), CXCL16 is primarily expressed in human cortical areas, including the orbitofrontal gyrus, retrosplenial cortex, and piriform cortex, which are involved in decision-making and olfactory processing. In the human hippocampus, CXCL16 expression is estimated at 24.5 normalized transcripts per million (nTPM), but its spatial distribution remains unexplored. In contrast, the Human Protein Atlas reports lower CXCL16 concentrations (0.5 nTPM) in the mouse hippocampus. Other studies have detected Cxcl16 RNA in hippocampal myeloid cells [35] and endothelial cells [19] in mice. Given the growing importance of CXCL16, this study investigates its expression in the rat hippocampus and neurons, a topic that has received limited attention.

Interestingly, CXCL16 was also expressed in Purkinje cells, particularly in their perikarya. This specific localization suggests a possible role in intracellular signaling rather than intercellular communication. Unexpectedly, PAE also altered the expression of NeuN in Purkinje cells. NeuN, a postmitotic neuronal marker with a known role as a splicing regulator [36], has been associated with the regulation of seizure susceptibility and anxiety-like behaviors in NeuN knockout mice [37]. While it is often reported that NeuN expression decreases during pathological events, our results suggest a possible overexpression of NeuN in Purkinje cells, a phenomenon that has not been previously described [38].

Variations in the expression of chemokines in the brain triggered by PAE can have various clinical effects [39]. These include neurodevelopmental disorders leading to cognitive and behavioral deficits, learning disorders, attention deficits and impulse control problems. It can also contribute to neuroinflammation, which can exacerbate neurological and neurodegenerative diseases, predispose people to neuroimmune diseases and increase susceptibility to infections and autoimmune diseases. In addition, altered neuroimmune signaling due to chemokine variations can lead to chronic pain conditions and be associated with an increased risk of mental health conditions such as anxiety and depression, directly impacting quality of life. The implications of these findings for neuronal function, cell signaling and PAE-associated neuropathology need to be further investigated.

## 4. Materials and Methods

### 4.1. Animals

The experiments were conducted in accordance with the official Mexican standard, NOM-062-ZOO-1999, and the guidelines of the local Committee for Animal Welfare and Use. This project was approved by the Ethics Committee at the University Center of Health Sciences of the University of Guadalajara (CE-22-113).

Female Wistar rats (250–300 g, 30–40 days old) were acquired from the National Institute of Neurobiology of UNAM and housed in polycarbonate cages in groups of three until they were paired with a male (300 g) for reproduction. The rats were housed at room temperature (24 ± 1 °C), a humidity of 50 ± 5% and a 12 h light–dark cycle and had free access to food and water. Mating was monitored for five hours to ensure proper copulation, which is considered day 0 of gestation (GD0). Once the dams were pregnant, they were individually caged and randomly assigned to each experimental group: prenatal alcohol exposure (PAE), isocaloric treatment (ISO) and intact control (INT). During gestation, the dams were weighed daily by the experimenters, which served as a habituation phase.

### 4.2. Alcohol Exposure

To replicate the conditions of heavy alcohol consumption during pregnancy, a well-established protocol was used [40]. In this protocol, rats received ethanol (Sigma Aldrich, Lenexa, KA, USA) in NaCl 0.9 water solution (PiSA, Mexico) 20% (*w*/*v*) at a dose of 6 g/kg per day during gestation days (GD) 8–20 via intragastric gavage. The pair-fed group (ISO) received a 28% sucrose solution at a dose of 10.5 g/kg per day, equivalent to the same caloric intake as ethanol, while the untreated group received no treatment during gestation. Animals were culled if the litter had more than 10 pups to rule out discrepancies due to nutritional deficiencies. At weaning in PD21, pups were divided into groups of 4–5 animals to avoid overcrowding and allocated into their subgroups according to sacrifice day: PD21, PD38, PD45 and PD60 and treatment: alcohol exposure (EtOH), isocaloric treatment (ISO) or intact control (INT) as shown below (Figure 10).

### 4.3. Processing of the Tissue

Both male and female pups at 21, 38, 45 and 60 days of age were used, with three rats per group for each technique. Each rat was weighed individually and anesthetized with a peritoneal injection of pentobarbital (50 mg/kg) and sacrificed by decapitation. To reduce the number of rats required for the study, the brains were divided into two hemispheres, with the right hemisphere being used for immunohistochemistry and the left hemisphere for subsequent molecular studies. After the right hemisphere was extracted and sliced with a stainless-steel matrix, it was fixed in 10% neutral buffered formalin for 24 h, embedded in paraffin for preservation, and cut into 3 μm sections using a microtome (Leica Biosystems RM2265 Automated), which were mounted on pre-loaded slides. To account for correlations between the hippocampal subfields and to minimize the number of univariate comparisons, group differences were analyzed using a repeated-measures ANOVA based on group averages (n = 3).

### 4.4. Immunohistochemistry

Tissue sections were routinely processed by heating and removing excess paraffin with xylene, followed by rehydration of the tissue in an alcohol series. After rehydration of the tissue sections, heat-induced epitope retrieval was achieved by immersion in a bath containing 10 mM sodium citrate solution (pH:6.0). Endogenous peroxidase activity was neutralized with 3% H_2_O_2_ for 10 min followed by unspecific protein blocking for 30 min. The tissue sections were then incubated overnight at 4 °C with the primary antibody anti-CXCL16 (#Cat. PA5-102391; dilution 1:500; Invitrogen, Carlsbad, CA, USA). Detection of the primary antibody was performed using the HiDef DetectionTM HRP Polymer System (Cat: 954D; Cell Marque, Rocklin, CA, USA) according to the manufacturer’s instructions. Sections were incubated with the substrate/chromogen 3,3′-diaminobenzidine (DAB) for 10 min and then counterstained with hematoxylin, dehydrated and mounted. Negative control tissues for CXCL16 were performed in both brain and testis, while the positive control was performed only in testis, where constitutive expression of CXCL16 is high [21]. Standardization of the technique included evaluation of IHC by a pathologist to ensure that there was no false negative signal.

### 4.5. Immunofluorescence

We relied on a previous optimization of the technique for paraffined embedded tissue sections [41] and our current protocol for IHC described previously. Briefly, the protocol followed the same steps as for IHC until incubation of the primary antibodies, which had different concentrations: CXCL16 polyclonal antibody (PA5-115068) from Thermo Fisher Scientific (formerly Invitrogen), Waltham, MA, USA, at a concentration of 1:200; the NeuN monoclonal antibody 1B7 (MA5-33103) from Thermo Fisher Scientific (formerly Invitrogen), Waltham, MA, USA, at a concentration of 1:500; the monoclonal GFAP antibody ASTRO6 (MA5-12023) at a concentration of 1:500; and the monoclonal IBA1 antibody GT10312 (MA5-27726) from Thermo Fisher Scientific (formerly Invitrogen), Waltham, MA, USA, at a concentration of 1:250. The tissues were then incubated for one hour in a dark, light-protected container protected from light with donkey anti-mouse IgG (H+L) ReadyProblesTM, Alexa FluorTM 488 (R37114) from Thermo Fisher, Waltham, MA, USA, to stain for cell type and goat anti-rabbit IgG (H+L) ReadyProbesTM, Alexa FluorTM 594 (R37117) from Thermo Fisher, Waltham, MA, USA, to stain for the presence of CXCL16. They were then rinsed with distilled water and mounted with Slowfade Diamond Antifade (ThermoFisher, Waltham, MA, USA). For immunofluorescence, the specificity of the staining was validated using a positive control test in rat testis, which confirmed CXCL16 expression in spermatogonia and spermatids (Appendix A).

### 4.6. Image Detection

Tissues from immunohistochemistry were scanned using the Aperio LV1 IVD (Leica Biosystems) and then quantified using Qupath v0.4.3 software. In the case of immunofluorescence, images were captured using a Carl Zeiss Axio Imager A2 fluorescence microscope and analyzed using Zeiss Zen v3.8 software.

### 4.7. QuPath Analysis

Tissue section images were analyzed using QuPath version 0.2.3, open-source software for digital pathology and whole-slide image analysis described by Bankhead et al. [42]. The software was developed using Java 8 and has a JavaFX interface for annotation and visualization; built-in algorithms for common tasks, including cell and tissue detection; and interactive machine learning for object and pixel classification. It is compatible with ImageJ (v1.54p), OpenCV (v4.11.0), Java Topology Suite (v1.20.0) and OMERO (v5.6.14). The software supports various image formats via Bio-Formats and OpenSlide, including whole slides.

### 4.8. Data Analysis

Data analysis was performed using GraphPad Prism v.10 software (San Diego, CA, USA) and RStudio 2023.12.1 (Posit Software, Boston, MA, USA). The Shapiro–Wilk test confirmed the normality of the data and allowed for parametric or non-parametric statistical analysis. Two-way ANOVA identified differences between groups with subsequent Tukey post-hoc tests. Results were expressed as mean ± standard error of the mean (SEM). Values of *p* < 0.05 were considered statistically significant.

## 5. Conclusions

Prenatal alcohol exposure (PAE) leads to maternal immune activation that disrupts fetal neurodevelopment. This common mechanism is also suspected in autism and other mental disorders, which affect a significant proportion of children worldwide. The differences in the distribution of CXCL16 expression observed in this study may provide insights into the role of chemokines in neuroplasticity in the hippocampus and potentially open new avenues for better treatment of FASD.

Possible approaches include the development of new neuroprotective agents using CXCL16 or its analogs for PAE-induced damage. In addition, the targeted use of chemokines such as CXCL16 in the context of anti-inflammatory treatments could help to mitigate the neurodevelopmental deficits associated with FASD. Future treatments should combine personalized medicine based on individual responses and biomarkers and could incorporate CXCL16-targeted therapies. Integrated approaches, including pharmacotherapy, nutritional supplementation and behavioral interventions, are critical to address the complex needs of individuals with FASD.

Understanding the distribution of this relatively understudied chemokine in the brain is critical to determine how PAE affects chemokine signaling during development and how these effects differ between the sexes. Notably, most studies disproportionately focus on males. Therefore, we focused on studying immune responses in females to address this gap.

A limitation of this study is that we measured the positivity of CXCL16 expression but not the intensity of expression or the downstream signaling pathways affected by CXCL16. In addition, employing scans of immunohistochemical slides lacks the precision offered by stereological methods. Therefore, a more in-depth analysis based on the current results is required for subsequent studies to accurately determine the local differences in chemokine expression. Nonetheless, our results provide a basis for future research to investigate the role and function of CXCL16 in the developing brain and to support efforts to better understand its involvement in neurodevelopmental processes. Future research could focus on early detection methods and preventive treatments that target chemokines such as CXCL16 to reduce the impact of PAE on neurodevelopment. These strategies emphasize the importance of ongoing research and development to support people with FASD.

## Figures and Tables

**Figure 1 ijms-26-01920-f001:**
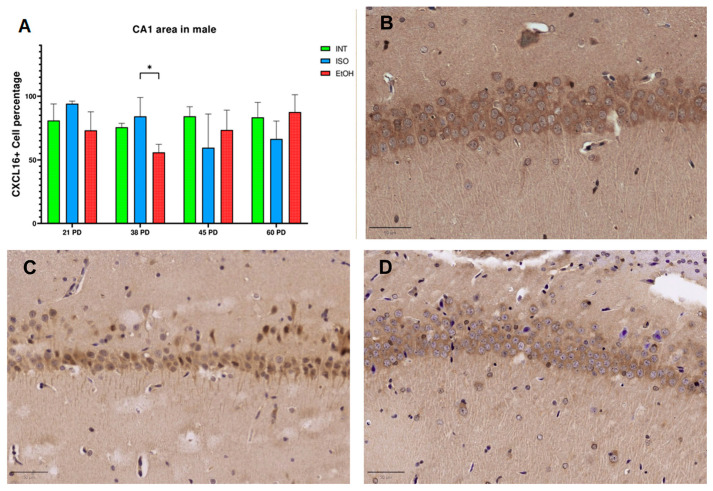
Expression of CXCL16 in male CA1 at PD 38. (**A**) Percentage of CXCL16+ cells by QuPath analysis and CXCL16 IHC staining in groups (**B**) EtOH, (**C**) INT and (**D**) ISO. PD: postnatal day. * *p* < 0.05. ISO: isocaloric, INT: intact, EtOH: alcohol exposure group. The bars indicate 50 μm.

**Figure 2 ijms-26-01920-f002:**
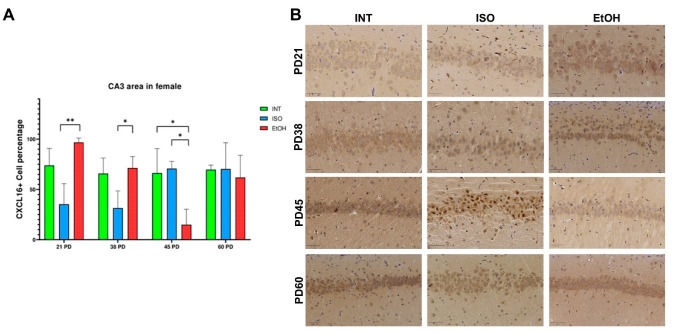
Expression of CXCL16 in female CA3. (**A**) Percentage of CXCL16+ cells by QuPath analysis. (**B**) CXCL16 distribution within groups and age of females in CA3. PD: postnatal day. * *p* < 0.05, ** *p* < 0.005, ISO: isocaloric, INT: intact, EtOH: alcohol exposure group. The bars indicate 50 μm.

**Figure 3 ijms-26-01920-f003:**
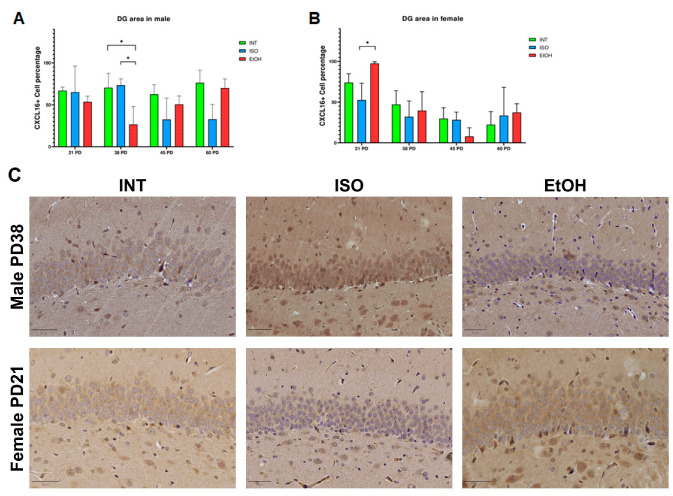
Expression of CXCL16 in male and female DGs. QuPath analysis of CXCL16+ cells per treatment in males (**A**) and females (**B**). Photomicrographs of groups of interest (**C**). PD: postnatal day. * *p* < 0.05. ISO: isocaloric, INT: intact, EtOH: alcohol exposure group, DG: dentate gyrus. The bars indicate 50 μm.

**Figure 4 ijms-26-01920-f004:**
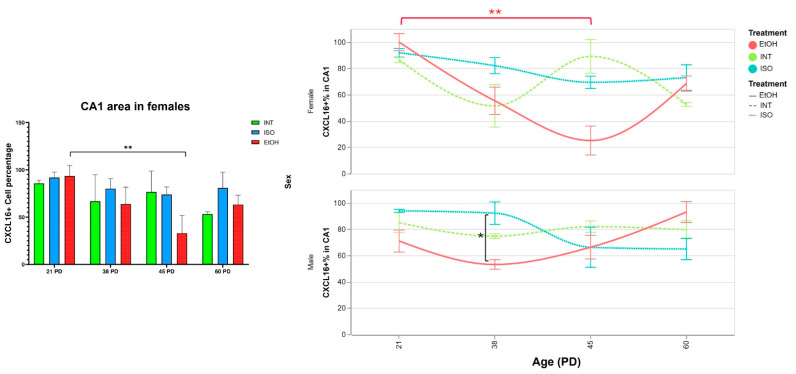
Expression of CXCL16 in CA1 throughout development. (**Left panel**) Percentage of CXCL16+ cells by QuPath analysis. (**Right panel**) Timeline showing CXCL16 variations over time in both females and males. Red brackets show temporal group comparisons within the EtOH group and black brackets represent differences between treatment groups shown in the previous comparisons. PD: postnatal day. * *p* < 0.05, ** *p* < 0.005. ISO: isocaloric, INT: intact, EtOH: alcohol exposure group.

**Figure 5 ijms-26-01920-f005:**
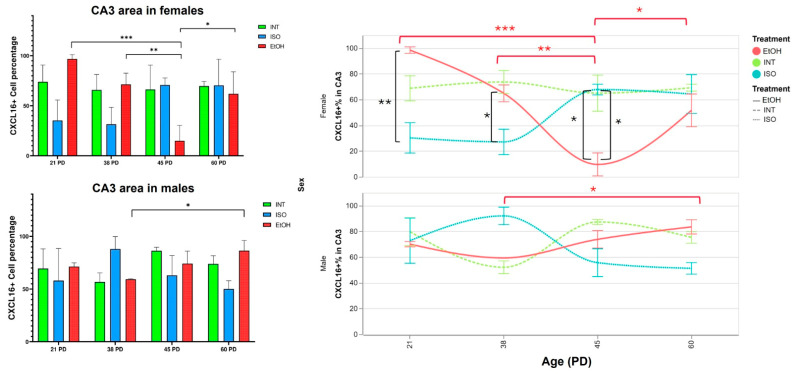
Expression of CXCL16 in CA3 through development. (**Left panel**) Percentage of CXCL16+ cells by QuPath analysis in females and males. (**Right panel**) Timeline showing CXCL16 changes over time in both females and males; note that the AMRR effect is present in the female EtOH group. Red brackets show group comparisons over time within the EtOH group and black brackets represent differences between treatment groups shown in previous comparisons. AMRR: Alcohol-Mediated Rebound Response, PD: postnatal day. * *p* < 0.05, ** *p* < 0.005, *** *p* < 0.001. ISO: isocaloric, INT: intact, EtOH: alcohol exposure group.

**Figure 6 ijms-26-01920-f006:**
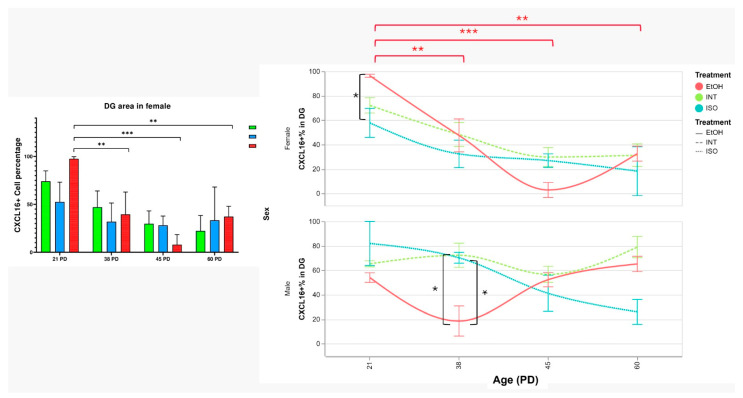
Expression of CXCL16 in female DG through development (**Left panel**) Percentage of CXCL16+ cells by QuPath analysis (**Right panel**) Timeline showing CXCL16 variations through time in both females and males. Red brackets show temporal group comparisons within the EtOH group and black brackets represent differences between treatment groups shown in previous comparisons. AMRR: Alcohol-Mediated Rebound Response, PD: postnatal day. * *p* < 0.05, ** *p* < 0.005, *** *p* < 0.001. ISO: isocaloric, INT: intact, EtOH: alcohol exposure group, DG: dentate gyrus.

**Figure 7 ijms-26-01920-f007:**
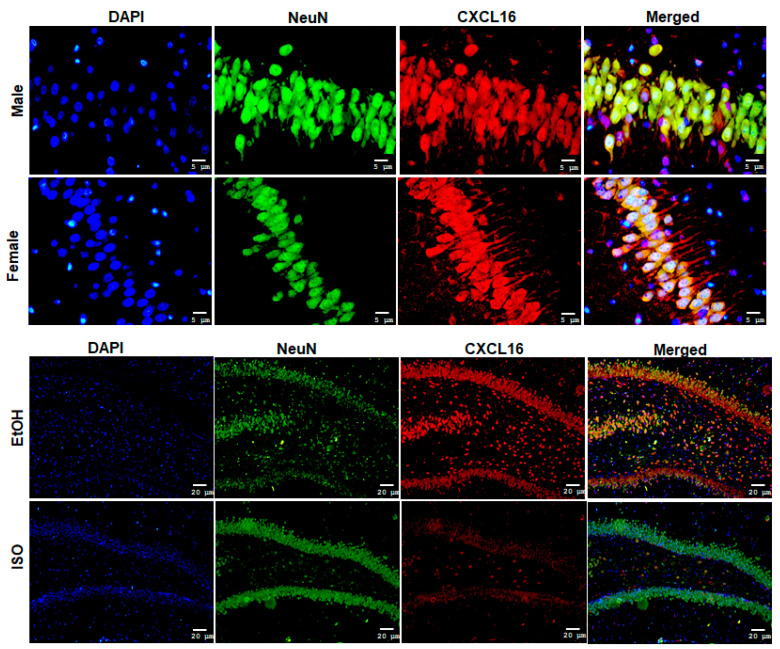
CXCL16 expression in DG neurons. (**Upper panel**) Double staining of CXCL16 and NeuN in the DG showing differences in the subcellular localization of CXCL16 in males and females exposed to EtOH at PD21 (40×) and (**lower panel**) spatial distribution of CXCL16 in neurons showed by females in ISO and EtOH groups (10×). The bars indicate 5 μm or 20 μm.

**Figure 8 ijms-26-01920-f008:**
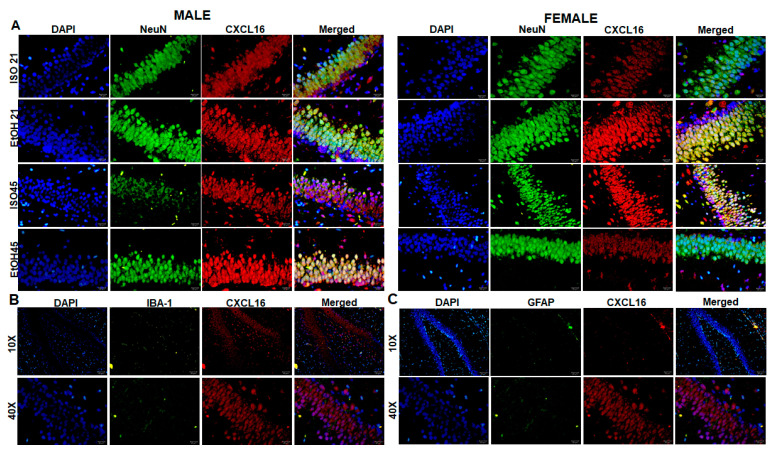
CXCL16 colocalization in neurons in DG. (**A**) Double staining of CXCL16 and NeuN among ISO and EtOH groups at PD21 and PD45 in males (**left**) and females (**right**). (**B**) Double staining of CXCL16 and microglia (IBA-1) and (**C**) double staining of CXCL16 and astrocyte marker (GFAP). The bars indicate 5 μm or 20 μm.

**Figure 9 ijms-26-01920-f009:**
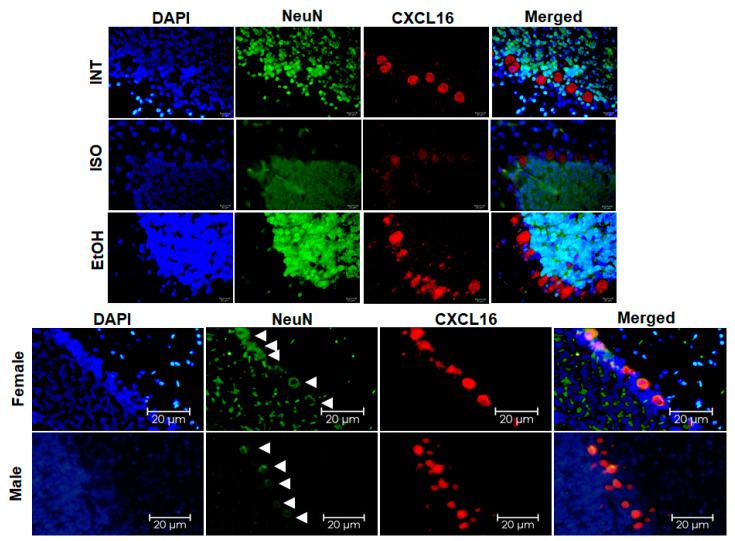
CXCL16 expression in Purkinje cells. (**Upper panel**) Double staining of CXCL16 and NeuN in female cerebellum at PD21. (**Lower panel**) Expression of CXCL16 among Purkinje cells of male and female from the EtOH group at PD21, with the presence of NeuN in these cells shown with arrows. The bars indicate 20 μm.

**Figure 10 ijms-26-01920-f010:**
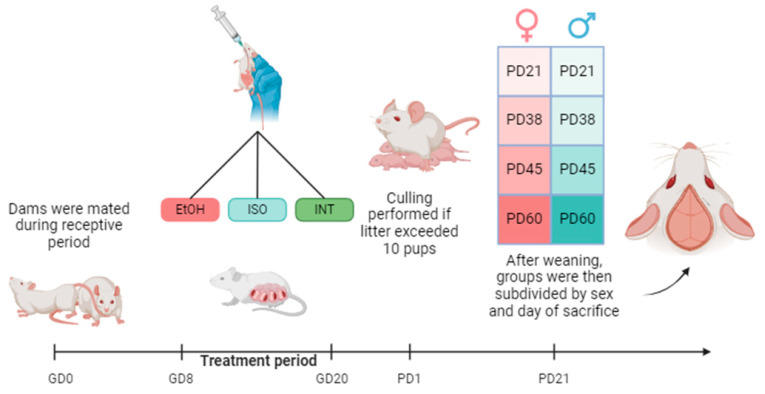
Experimental design. Created in BioRender.com.

## Data Availability

The data presented in this study are available on request from the corresponding author.

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
