# Peer review of "Prenatal Alcohol Exposure Disrupts CXCL16 Expression in Rat Hippocampus: Temporal and Sex Differences"

_ijms, 2025, doi:10.3390/ijms26051920_

Round 1

Reviewer 1 Report

Comments and Suggestions for Authors

I have only a few requests for this otherwise excellent manuscript.

First, the authors should point out that using scans of immunohistochemical slides is not as precise as stereology would be.

Second, the authors should discuss more fully the range of measures that led to the decision to analyze group differences, and the implication of this for clinical knowledge with respect to FAE.

Author Response

Reviewer #1

I have only a few requests for this otherwise excellent manuscript.

ANSWER: We would like to thank the reviewer for his positive comments on our work and for the time taken to review our manuscript in such detail. Below you will find the answers to the reviewer's queries.

  • First, the authors should point out that using scans of immunohistochemical slides is not as precise as stereology would be.

Answer: We thank the reviewer for pointing out this important aspect. In accordance with this, we have added the following as a limitation to the work: "In addition, employing scans of immunohistochemical slides lacks the precision offered by stereologic methods. Therefore, a more in-depth analysis based on the current results is required for subsequent studies to accurately determine the differences in chemokine expression."

  • Second, the authors should discuss more fully the range of measures that led to the decision to analyze group differences, and the implication of this for clinical knowledge with respect to FAE.

Answer: We thank the reviewer for the opportunity to further explain this relevant clinical aspect as follows "Variations in the expression of chemokines in the brain triggered by PAE can have various clinical effects [40]. These include neurodevelopmental disorders leading to cognitive and behavioral deficits, learning disorders, attention deficits and impulse control problems. It can also contribute to neuroinflammation, which can exacerbate neurological and neurodegenerative diseases, predispose people to neuroimmune diseases and increase susceptibility to infections and autoimmune diseases. In addition, altered neuroimmune signaling due to chemokine variations can lead to chronic pain conditions and be associated with an increased risk of mental health conditions such as anxiety and depression, directly impacting quality of life

Reviewer 2 Report

Comments and Suggestions for Authors

The article presents a study on the effects of prenatal alcohol exposure (PAE) on chemokine signaling, with a specific focus on CXCL16 expression in the developing brain.

Detailed comments and suggestions for improvement are provided below:

  1. The introduction should be more explicit regarding the specific aims of the study. What is/are the purpose(s) of this research?
  2. To investigate the expression of CXCL16 in the brain following PAE?
  3. To examine the spatial distribution of CXCL16 protein in the brain?

iii. To explore developmental changes in CXCL16 expression over time?

  1. To determine whether sexual dimorphism affects CXCL16 expression after PAE?
  2. Lines 91-96: Consider moving this text to the data analysis section.
  3. In the prenatal administration protocol, the control group received a solution of sucrose. It is suggested sucrose impacts the reward circuit, including the nucleus accumbens. Authors should consider that Cxcl16 regulation in this area may be due to prenatal alcohol exposure, not sucrose.
  4. There is a lack of discussion regarding how the results align or contrast with those of other studies. For instance, the authors could address whether the distribution of CXCL16 expression in their mouse model during prenatal alcohol exposure is consistent with, or different from, similar studies.
  5. While the authors mention that the findings could 'open new possibilities' for the treatment of fetal alcohol spectrum disorders (FASD), specific examples of how these findings might influence therapeutic strategies or future treatments for FASD should be included.
  6. Please discuss the limitations of the presented study and suggest potential avenues for future research or improvements.
Comments on the Quality of English Language

A light editing can be useful

Author Response

Reviewer #2

Detailed comments and suggestions for improvement are provided below:

ANSWER: We would like to thank the reviewer for the positive comments on our work and for the time taken to review our manuscript in such detail. Undoubtedly the comments have improved our work. Below you will find the answers to the reviewer's queries.

  • The introduction should be more explicit regarding the specific aims of the study. What is/are the purpose(s) of this research?
    1. To investigate the expression of CXCL16 in the brain following PAE?
    2. To examine the spatial distribution of CXCL16 protein in the brain?
    3. To explore developmental changes in CXCL16 expression over time?
    4. To determine whether sexual dimorphism affects CXCL16 expression after PAE?

Answer: We thank the reviewer for bringing to our attention this important aspect. Consequently, we included more explicitly our aims as follows: "Therefore, the aim of this study is to investigate the expression patterns of CXCL16 in the brain after prenatal alcohol exposure (PAE). In particular, this research aims to investigate the spatial distribution of the chemokine, assess the influence of sexual dimorphism on its expression, and explore potential developmental changes over time. These results will improve our understanding of the presence and role of CXCL16 in the brain."

  • Lines 91-96: Consider moving this text to the data analysis section.

Answer: Done, the text was moved to sections 4.3 and 4.5 were it corresponds.

  • In the prenatal administration protocol, the control group received a solution of sucrose. It is suggested sucrose impacts the reward circuit, including the nucleus accumbens. Authors should consider that Cxcl16 regulation in this area may be due to prenatal alcohol exposure, not sucrose.

Answer: This is a very pertinent question. We thank the reviewer for the opportunity to address this aspect. We have therefore reworded a paragraph as follows: "It is important to note that the ISO and INT groups did not behave completely the same, as the observed differences between the EtOH-INT and EtOH-ISO comparisons were slightly different. This is an interesting fact considering that most studies on PAE only include an ISO group and no intact group as reference. This discrepancy could be related to the effects that sucrose consumption could have on some areas of the brain. In the hippocampus, for example, sucrose consumption has been associated with impaired spatial memory and increased neuroinflammatory markers such as IL-6 and IL-1β [22] and could also affect neurogenesis in a similar way to early childhood stress exposure [23]. However, in the case of CXCL16, the changes in the regulation of Cxcl16 expression in the brain are due to prenatal alcohol exposure rather than sucrose, as shown by our results here as well as those previously described [20].”

  • There is a lack of discussion regarding how the results align or contrast with those of other studies. For instance, the authors could address whether the distribution of CXCL16 expression in their mouse model during prenatal alcohol exposure is consistent with, or different from, similar studies.

Answer: We thank the reviewer for the opportunity to elaborate on this aspect. With this in mind, we have added the following information from three related references on CXCL16 expression in mouse and human brain: "According to the Human Protein Atlas (available at v19.proteinatlas.org), CXCL16 is primarily expressed in human cortical areas, including the orbitofrontal gyrus, retrosplenial cortex, and piriform cortex, which are involved in decision-making and olfactory processing. In the human hippocampus, CXCL16 expression is estimated at 24.5 normalized transcripts per million (nTPM), but its spatial distribution remains unexplored. In contrast, the Human Protein Atlas reports lower CXCL16 concentrations (0.5 nTPM) in the mouse hippocampus. Other studies have detected Cxcl16 RNA in hippocampal myeloid cells (Rosen et. al., 2022) and endothelial cells (Siqueira et. al., 2022) in mice. Given the growing importance of CXCL16, this study investigates its expression in the rat hippocampus and neurons, a topic that has received limited attention."

  • While the authors mention that the findings could 'open new possibilities' for the treatment of fetal alcohol spectrum disorders (FASD), specific examples of how these findings might influence therapeutic strategies or future treatments for FASD should be included.

Answer: We thank the reviewer for the opportunity to address this aspect in more detail. We therefore develop the explanation of the open possibilities as follows: "Possible approaches include the development of new neuroprotective agents using CXCL16 or its analogs for PAE-induced damage. In addition, the targeted use of chemokines such as CXCL16 in the context of anti-inflammatory treatments could help to mitigate the neurodevelopmental deficits associated with FASD. Future treatments should combine personalized medicine based on individual responses and biomarkers and incorporate CXCL16-targeted therapies. Integrated approaches, including pharmacotherapy, nutritional supplementation, and behavioral interventions, are critical to address the complex needs of individuals with FASD"

  • Please discuss the limitations of the presented study and suggest potential avenues for future research or improvements.

Answer: We thank the reviewer for the opportunity to further elaborate into this aspect. We included limitations and perspectives as follows: "A limitation of this study is that we measured the positivity of CXCL16 expression, but not the intensity of expression or the downstream signaling pathways affected by CXCL16. In addition, employing scans of immunohistochemical slides lacks the precision offered by stereological methods. Therefore, a more in-depth analysis based on the current results is required for subsequent studies to accurately determine the local differences in chemokine ex-pression. Nonetheless, our results provide a basis for future research to investigate the role and function of CXCL16 in the developing brain and to support efforts to better under-stand its involvement in neurodevelopmental processes. Future research could focus on early detection methods and preventive treatments that target chemokines such as CXCL16 to reduce the impact of PAE on neurodevelopment. These strategies emphasize the importance of ongoing research and development to support people with FASD."
